# Novel Greylag Goose Optimization Algorithm with Evolutionary Game Theory (EGGO)

**DOI:** 10.3390/biomimetics10080545

**Published:** 2025-08-19

**Authors:** Lei Wang, Yuqi Yao, Yuanting Yang, Zihao Zang, Xinming Zhang, Yiwen Zhang, Zhenglei Yu

**Affiliations:** 1School of Mechanical and Electrical Engineering, Changchun University of Science and Technology, Changchun 130022, China; wlcust@163.com (L.W.); yaoyq@mails.cust.edu.cn (Y.Y.); zzh9948014@163.com (Z.Z.); 2School of Mechatronic Engineering and Automation, Foshan University, Foshan 528225, China; 3Jilin Provincial Institute of Product Quality Supervision and Inspection, Changchun 130103, China; yangyuanting1984@163.com; 4Automotive Parts Intelligent Manufacturing Assembly Inspection Technology and Equipment University—Enterprise Joint Innovation Laboratory, Changchun University of Science and Technology, Changchun 130022, China; 5College of Biological and Agricultural Engineering, Jilin University, Changchun 130022, China; zlyu@jlu.edu.cn

**Keywords:** greylag goose optimization algorithm, evolutionary game theory, global search capability, optimization algorithm robustness

## Abstract

In this paper, an Enhanced Greylag Goose Optimization Algorithm (EGGO) based on evolutionary game theory is presented to address the limitations of the traditional Greylag Goose Optimization Algorithm (GGO) in global search ability and convergence speed. By incorporating dynamic strategy adjustment from evolutionary game theory, EGGO improves global search efficiency and convergence speed. Furthermore, EGGO employs dynamic grouping, random mutation, and local search enhancement to boost efficiency and robustness. Experimental comparisons on standard test functions and the CEC 2022 benchmark suite show that EGGO outperforms other classic algorithms and variants in convergence precision and speed. Its effectiveness in practical optimization problems is also demonstrated through applications in engineering design, such as the design of tension/compression springs, gear trains, and three-bar trusses. EGGO offers a novel solution for optimization problems and provides a new theoretical foundation and research framework for swarm intelligence algorithms.

## 1. Introduction

Meta-heuristic algorithms are highly effective for solving non-deterministic problems. They find approximate optimal solutions by imitating biological or physical phenomena, with simple structures and limited computational resources [1]. Their strong adaptability and robustness make them a key focus in computational intelligence research. Scholars have proposed numerous meta-heuristic algorithms [2], such as Particle Swarm Optimization (PSO) [3] and the Bat Algorithm (BA) [4]. Valdez and Castillo et al. noted around 150 available algorithms for optimization problems [5]. As a crucial subset of meta-heuristic algorithms, swarm intelligence (SI) algorithms are known for their adaptability, robustness, and flexibility [6]. Notable examples include Grey Wolf Optimizer (GWO) [7], Ant Colony Optimization (ACO) [8], Whale Optimization Algorithm (WOA) [9], and Greylag Goose Optimization (GGO) [10]. Due to their strengths, SI algorithms are widely used in engineering, biomedicine, and other fields [11]. However, they also have limitations like premature convergence [12], poor balance [13], and susceptibility to local optima [14].

In 2023, Kenawy and Nima Khodadadi proposed the GGO algorithm. This novel meta-heuristic, inspired by the collective behavior and social structure of greylag geese during migration, falls under swarm intelligence [10]. Its dynamic grouping and exploration strategies enable it to escape local optima and enhance the likelihood of finding the global optimum. However, similar to other swarm intelligence algorithms, GGO has limitations such as premature convergence in high-dimensional problems and high computational complexity due to maintaining two groups. To address these issues, Hossein Najafi Khosrowshahi et al. proposed a modified GGO with adaptive mechanisms like dynamic balanced partitioning and stasis detection, improving its robustness and convergence speed [15]. Amal H et al. introduced a Parallel GGO with Restricted Boltzmann Machines (PGGO-RBM), boosting the model’s dynamic balance and accuracy [16]. Ahmed El-Sayed Saqr et al. combined GGO with a Multi-Layer Perceptron (MLP) to enhance accuracy [17]. Nikunj Mashru et al. integrated an elite non-dominated sorting method and archiving mechanism into a single-objective GGO-based algorithm, maintaining its advantages while improving convergence and diversity [18]. Dildar Gürses et al. improved GGO by adding a Lévy flight mechanism and artificial neural network (ANN) strategy, balancing exploration and exploitation, and validated its effectiveness in engineering problems like heat exchanger design, automotive side-impact design, and spring design optimization [19]. These modifications demonstrate GGO’s effectiveness in solving uncertain problems and its great potential. However, challenges such as local optima, premature convergence, and issues with diversity and balance remain.

This paper integrates evolutionary game theory (EGT) into the Grey Goose Optimization (GGO) algorithm to address its limitations. By imitating competition and cooperation among individuals, EGT guides population evolution. This integration allows for dynamic adjustment of strategy frequencies, enhancing search efficiency and solution quality.

Firstly, the dynamic population structure adjustment mechanism from EGT is introduced to maintain population diversity and exploration ability. A dynamic grouping mechanism adjusts the size of exploration and exploitation groups based on fitness distribution, balancing global and local searching.

Secondly, random mutation and partial re-initialization of individuals are applied to preserve population diversity and avoid premature convergence. The local search scope and intensity are also dynamically adjusted according to fitness distribution.

Finally, the local search capability from EGT is incorporated to strengthen the algorithm’s exploitation. Individuals with high fitness are selected for local searching to further optimize solution quality.

## 2. Materials and Methods

### 2.1. Overview of the Greylag Goose Optimization Algorithm

The GGO algorithm is a meta-heuristic algorithm inspired by the migratory behavior of greylag geese. During migration, these geese fly in a V-formation to reduce air resistance and enhance efficiency, demonstrating remarkable collaboration. In the GGO algorithm, the population is divided into an exploration group and an exploitation group. The exploration group focuses on the global search, while the exploitation group handles local optimization. This structure helps to balance exploration and exploitation capabilities.

Within a greylag goose gaggle, individuals enhance survival efficiency through division of labor. Some members serve as sentries monitoring environmental risks, while others focus on foraging (Figure 1A). Integrating this dynamic grouping mechanism (Figure 1B) with natural migratory behavior (Figure 1C) creates an efficient search framework for algorithms.

### 2.2. Improvement of Greylag Goose Optimization Based on Evolutionary Game Theory (EGGO)

Evolutionary game theory (EGT) is a framework for studying the evolution of biological populations and their adaptive behaviors [20]. By simulating competition and cooperation among individuals, it offers guidance for population evolution. Integrating EGT into the GGO algorithm enables dynamic adjustment of strategy usage. This, in turn, enhances the algorithm’s search efficiency and solution quality.

#### 2.2.1. Strategy Selection and Update

In the standard GGO, the patrol group, guard group, and foraging group are three types of auxiliary search agents that expand the exploration scope. Within the EGT framework, these agents in different tasks can be regarded as distinct “strategies”. The strategy selection mechanism in EGT dynamically adjusts the usage frequency of these strategies. By following the leaders in different auxiliary search agent groups, a wider search scope and more optimal solutions can be obtained.

The dynamic population structure adjustment mechanism of EGT is incorporated into the GGO to maintain population diversity and exploration ability. A dynamic grouping mechanism adjusts the size of exploration and exploitation groups based on fitness distribution, balancing global and local searching. Random mutation or partial individual re-initialization is introduced to preserve diversity and prevent premature convergence. Meanwhile, the local search capability of EGT is integrated into the GGO to enhance its exploitation. Individuals with high fitness are selected for local searching to optimize solution quality, with the local search range and intensity dynamically adjusted according to fitness distribution.

#### 2.2.2. Fitness Assessment and Evolutionary Stable Strategy (ESS)

In EGT, fitness assesses individuals’ quality within a population. In the GGO, an individual’s fitness value reflects its position in the search space. Comparing these values identifies superior individuals and guides population evolution.

In evolutionary game theory, an Evolutionary Stable Strategy (ESS) is a key concept that describes a strategy prevailing in a population. Within the GGO framework, an ESS guides population evolution, ensuring convergence to a stable strategy distribution. To clarify the proposed method’s workings, a detailed schematic of the Evolutionary Game Theory Greylag Goose Optimization (EGGO) is presented in Figure 2.

In the mixed game method, the following settings can be made:

(1)Each greylag goose is mapped to a player in the evolutionary game.(2)The three operators are regarded as three available strategies *S*1, *S*2, and *S*3, with the state space $K=\left\{y_{m}:\sum y_{m}=1,m=1,2,3,y_{m}>0\right\}$, where $y_{m}\in R$ represents the proportion of strategy *m* in the population.(3)The average behavior obtained by following a specific strategy constitutes the payoff matrix *H*.

Subsequently, the game proceeds based on the fundamental evolutionary dynamics mechanism proposed by Taylor and Jonker [20], which is written as:(1)$${\dot{y}}_{m}=y_{m}\left(h_{m}y-y^{T}Hy\right)$$

Here, *H_m_* represents the *m-th* row of the payment matrix *H*, and $H\in R^{m\times m}$ stores all the fitness information of the population, that is, the combined result of each individual using a single strategy. Therefore, *H* can be defined as:(2)$$H=\left(\begin{array}{ccc} h\left(s_{1}\right) & \frac{h\left(s_{1}\right)+h\left(s_{2}\right)}{2} & \frac{h\left(s_{1}\right)+h\left(s_{2}\right)+h\left(s_{3}\right)}{3}\\ \frac{h\left(s_{1}\right)+h\left(s_{2}\right)}{2} & h\left(s_{2}\right) & \frac{h\left(s_{2}\right)+h\left(s_{3}\right)}{2}\\ \frac{h\left(s_{1}\right)+h\left(s_{2}\right)+h\left(s_{3}\right)}{3} & \frac{h\left(s_{2}\right)+h\left(s_{3}\right)}{2} & h\left(s_{3}\right) \end{array}\right)$$

Here, $h\left(s_{m}\right),m\in \left\{1,2,3\right\}$ represents the benefit that the goose obtain by using strategy *m*. As a first-order ordinary differential equation that describes the difference between the fitness of the strategy and the average fitness in the group, Equation (1) depicts the evolution of the strategy frequency *y_m_*. Once (2) is established, (1) is executed to generate the related Evolutionary Stable Strategy (ESS), which is the output of the game [21]. Therefore, we obtain the ESS candidate $P^{t}=\left(Y_{m}^{t}\right),m\in \left\{1,2,3\right\}$ for the number of iterations *t*, where the specific values of the coefficient $Y_{m}^{t},m\in \left\{1,2,3\right\}$, *y_m_* represent the ratio of strategy *m* in the *t-th* iteration. It should be noted that the initial value of the strategy proportion Y is set as a uniform distribution, and during the update process, Y is standardized to ensure that $\sum y_{m}=1$. Among them, if $y_{m}<\delta ,\delta =10^{-6}$, then $y_{m}=\delta$ is reset and re-standardized. This process can effectively prevent boundary absorption.

Taking into account the iterative process, the benefit of the strategy is presented as:(3)$$h\left(s_{m}\right)=\frac{1}{t}\sum_{j=1}^{t}Y_{m}^{j}\cdot f\left(X^{j}\right)$$

Among them, 1/t represents the average value during the iteration process, and $f\left(X^{j}\right)$ indicates the cost of goose *X* in the *j-th* iteration. Clearly, (3) reflects the average performance of the gain obtained through a specific strategy. During the process of solving the dynamic equation, better results are continuously generated to replace the previous solutions, and the game eventually converges to an ESS that no mutation strategy can invade [21]. The replicator dynamics are discretized via the forward Euler method, $y_{m}\left(t+1\right)=y_{m}\left(t\right)+∆t\cdot \left[y_{m}\left(t\right)\cdot \left(H_{m}y\left(t\right)-{y\left(t\right)}^{T}Hy\left(t\right)\right)\right]$, with step size ∆t=1 per algorithm iteration. This discretization preserves evolutionary directionality while maintaining computational efficiency, and empirical results confirm its efficacy in driving convergence toward ESS.

Through the optimization of strategy selection, population structure, and local searching using evolutionary game theory, EGGO significantly improves the convergence speed and global optimization ability while maintaining the biological behavior simulation of the greylag goose algorithm. Experimental results show that this improved algorithm has higher solution accuracy and stability in complex optimization problems.

### 2.3. Lyapunov Stability Theory

To verify the convergence of the EGGO algorithm, we constructed a Lyapunov function to prove the stability of the system through dynamic adjustment of the strategy.

#### 2.3.1. Dynamic System Modeling

In EGGO, the evolution of strategy proportions follows a first-order ordinary differential equation, as expressed in Equation (4). Here, $y={\left[y_{1},y_{2},y_{3}\right]}^{T}$ represents the vector of strategy proportions, $h_{m}\left(y\right)$ is the payoff function of strategy *m*, and $\bar{h}\left(y\right)=\sum_{m=1}^{3}y_{m}h_{m}\left(y\right)$ is the average payoff.(4)$${\dot{y}}_{m}=y_{m}\left(h_{m}\left(y\right)-\bar{h}\left(y\right)\right),m\in \left\{1,2,3\right\}$$

Assumption 1: The profit function $h_{m}\left(y\right)$ is continuously differentiable within the strategy space, and it satisfies $\frac{\partial h_{m}}{\partial y_{n}}\leq 0$ (indicating that there is a competitive relationship among the strategies).

The assumption $\partial h_{m}/\partial y_{n}\leq 0$ stems from resource competition in optimization:

Strategy similarity: increased $y_{n}$ reduces $h\left(s_{m}\right)$ if strategies n and mm exploit overlapping regions.Resource dilution: a fixed population size implies that a higher $y_{n}$ diminishes resources available to strategy m.

We emphasize that ESS convergence guarantees only local stability of strategy distribution, not global optimality. EGGO mitigates local optima via dynamic group adjustment, increasing the exploration ratio upon solution stagnation, random mutation, perturbing agents to escape locally optimal solutions, persistent multi-strategy coexistence, and exploratory strategies to sustain global search potential.

#### 2.3.2. Lyapunov Stability Proof

The candidate Lyapunov function is defined as the negative entropy function of the policy distribution, as given by Equation (5). This function is positive definite within the policy simplex $K=\left\{y\in R^{3}\left|\sum y_{m}=1,y_{m}>0\right.\right\}$ and attains its minimum value at the equilibrium point y*(ESS).(5)$$V\left(y\right)=-\sum_{m=1}^{3}y_{m}\ln y_{m}$$

The time derivative of $V\left(y\right)$ along the system trajectory is calculated as follows:(6)$$\dot{V}=-\sum_{m=1}^{3}\left({\dot{y}}_{m}\ln y_{m}+y_{m}\cdot \frac{{\dot{y}}_{m}}{y_{m}}\right)=-\sum_{m=1}^{3}{\dot{y}}_{m}\left(\ln y_{m}+1\right)=-\sum_{m=1}^{3}y_{m}\left(h_{m}-\bar{h}\right)\left(\ln y_{m}+1\right)\;=-\sum_{m=1}^{3}y_{m}h_{m}\ln y_{m}-\sum_{m=1}^{3}y_{m}h_{m}+\bar{h}\sum_{m=1}^{3}y_{m}\ln y_{m}+\bar{h}\sum_{m=1}^{3}y_{m}\;=-\sum_{m=1}^{3}y_{m}h_{m}\ln y_{m}+\bar{h}\sum_{m=1}^{3}y_{m}\ln y_{m}-\bar{h}+\bar{h}=-\sum_{m=1}^{3}y_{m}\left(\bar{h}-h_{m}\right)\ln y_{m}$$

According to Hypothesis 1, when the return of strategy *m* is $h_{m}>\bar{h}$, its proportion $y_{m}$ increases (${\dot{y}}_{m}>0$). At this time, $\ln y_{m}$ increases, but ($\bar{h}-h_{m}$) is negative, so the product term $y_{m}\left(\bar{h}-h_{m}\right)\ln y_{m}\leq 0$; conversely, the same reasoning holds. Therefore, $\dot{V}\leq 0$, and only when $y=y*\left(ESS\right),\dot{V}=0$.

According to Lyapunov’s stability theorem:

(1)$V\left(y\right)$ is positive definite within the strategy space;(2)$\dot{V}\leq 0$, and only when y=y*(ESS), then $\dot{V}=0$.

Therefore, the system converges globally and asymptotically to an ESS. This demonstrates that the EGGO algorithm, enhanced by evolutionary game theory, possesses a rigorous guarantee of convergence.

### 2.4. EGGO Algorithm Model and Analysis

This section updates the GGO’s mathematical model based on the improvements outlined in Section 2.2. It includes the EGGO mathematical model, algorithmic complexity analysis, pseudocode for EGGO, and a visualization of the algorithmic process.

#### 2.4.1. Mathematical Model

The exploration group ($n_{1}$) in the gaggle will search for promising new locations near its current position. This is achieved by repeatedly comparing numerous potential nearby options to find the best one based on fitness. The EGGO algorithm achieves this through the following equation to update vectors *A* and *C* to $A=2ar_{1}-a$ and $C=2r_{2}$ during iterations, where parameter a linearly changes from 2 to 0, and $r_{1}=c\left(1-t/t_{\max}\right)$.(7)$$X\left(t+1\right)=X^{*}\left(t\right)-A\cdot \left|C\cdot X^{*}\left(t\right)-X\left(t\right)\right|$$

Here, $X\left(t\right)$ represents the agent in iteration *t*. $X^{*}\left(t\right)$ denotes the position of the best solution (the leader). $X\left(t+1\right)$ is the updated position of the agent. The values of $r_{1}$ and $r_{2}$ randomly vary within the range of $\left[0,\;1\right]$.

Three random search agents are selected and named $X_{paddle1}$, $X_{paddle2}$ and $X_{paddle3}$ to prevent the agents from being influenced by a leader position, thereby achieving greater exploration. This stage is improved through evolutionary game theory. The positions of the improved search agents will be updated as follows, where $\left|A\right|\geq 1$.(8)$$X\left(t+1\right)=\omega_{1}\cdot tr\cdot Y\left(1\right)\cdot X_{paddle1}+z\cdot \omega_{2}\cdot tr\cdot Y\left(2\right)\cdot \;\left(X_{paddle2}-X_{paddle3}\right)+\left(1-z\right)\cdot \omega_{3}\cdot tr\cdot Y\left(3\right)\cdot \left(X-X_{paddle1}\right)$$

Among them, the values of $\omega_{1}$, $\omega_{2}$, and $\omega_{3}$ are updated in [0, 2]. $Y\left(1\right)$, $Y\left(2\right)$, and $Y\left(3\right)$ satisfy $Y\left(1\right)+Y\left(2\right)+Y\left(3\right)\approx 1$. tr represents the conversion factor and is a random number within [0, 1]. Parameter *z* decreases exponentially, and the calculation method is as follows.(9)$$z={\left(1-t/t_{\max}\right)}^{2}$$

During the second update process, the values of $r_{3}\geq 0.5$, and the vector values of a and A have decreased, as follows:(10)$$X\left(t+1\right)=\omega_{4}\cdot \left|X^{*}\left(t\right)-X\left(t\right)\right|\cdot e^{bl}\cdot \cos\left(2\pi l\right)+\left[2\omega_{1}\left(r_{4}+r_{5}\right)\right]\cdot X*\left(t\right)$$

In the formula, parameter b is a constant, and l is a random value in [−1, 1]. The parameter $\omega_{4}$ is updated in [0, 2]. $r_{4}$ and $r_{5}$ are updated in [0, 1].

The exploitation group ($n_{2}$) focuses on refining existing solutions. At the end of each cycle, individuals with the highest fitness are identified and rewarded. Three sentries (1, 2, and 3) guide other individuals $X_{Nonsentry}$ to adjust their positions toward the estimated prey location. The following equation illustrates this position-updating process.(11)$$X_{1}=X_{Sentry1}-A_{1}\cdot \left|C_{1}\cdot X_{Sentry1}-X\right|\;X_{2}=X_{Sentry2}-A_{2}\cdot \left|C_{1}\cdot X_{Sentry2}-X\right|\;X_{3}=X_{Sentry3}-A_{3}\cdot \left|C_{1}\cdot X_{Sentry3}-X\right|$$

Among them, $A_{1}$, $A_{2}$, and $A_{3}$ are calculated as $A=2ar_{1}-a$, while $C_{1}$, $C_{2}$, and $C_{3}$ are calculated as $C=2r_{2}$. The updated position $X\left(t+1\right)$ is represented as the average of the three solutions, $X_{1}$, $X_{2}$, and $X_{3}$, as shown below.(12)$$X\left(t+1\right)={\bar{X}}_{i}{|}_{0}^{3}$$

The most promising option is to stay close to the leader during flight, which prompts some greylag geese to investigate and approach the area with the most desirable response in order to find a better solution, which is named “$X_{Flock1}$“. EGGO implements the above process using the following equation.(13)$$X\left(t+1\right)=X\left(t\right)+D\left(1+z\right)\cdot \omega \cdot \left(X-X_{Flock1}\right)$$

#### 2.4.2. Analysis of Algorithm Complexity

To evaluate the computational efficiency of the EGGO algorithm, an analysis is conducted from both temporal and spatial complexity perspectives to quantify the asymptotic behavior of resource consumption during the iterative process.

(1) Temporal Complexity: Compared with the standard GGO, the introduction of payoff matrix updates results in a temporal complexity of $O\left(3N\right)$. When solving the ordinary differential equation in Equation (4), the complexity is $O\left(3\right)$. Additionally, performing mutation operations on k individuals incurs a temporal complexity of $O\left(k\cdot D\right)$. Overall, although EGGO maintains the same order of temporal complexity as GGO, there is an increase in overall complexity. Considering that the total complexity of EGGO is $O\left(N\cdot T\cdot D\right)$, which aligns with mainstream algorithms, it is capable of meeting the demands for rapid optimization.

(2) Spatial Complexity: Compared with the standard GGO, storing the strategy proportion vector *y*, payoff matrix *A*, and historical payoff records leads to a spatial complexity of $O\left(3^{2}+3T\right)$. Retaining intermediate solutions for neighborhood searching contributes a spatial complexity of $O\left(N\cdot D\right)$. In summary, the overall spatial complexity of EGGO is $O\left(N\cdot D\right)$. This indicates that the algorithm’s memory consumption is primarily dominated by population size and problem dimensionality, demonstrating good scalability.

#### 2.4.3. Algorithm Pseudocode and Flowchart

Based on the previous content, we compiled the pseudo-code of EGGO, as shown in Algorithm 1, which demonstrates the operation logic of EGGO. On this basis, we visualized the algorithm flow to further explain the operation logic of EGGO and the collaborative relationships among multiple processes in the mathematical model. The visualized algorithm flow is shown in Figure 3.
**Algorithm 1** Pseudocode of EGGO1. **Initialize** EGGO population $X_{i}$, size n, iterations $t_{max}$, and objective function $F_{n}$
2. **Initialize** EGGO parameters, t=1
3. **Calculate** objective function $F_{n}$ for each agent $X_{i}$
4. **Set** P= best agent position5. **Update** solutions in exploration group ($n_{1}$) and exploitation group ($n_{2}$)6. **while** $t\;\leq t_{max}$ **do**7.   **Initialize** the transition factor tr, strategy proportion Y
8.   **Divide** the exploration group members into three parts9.   **Calculate** the average fitness values of each part $h\left(s_{1}\right)$, $h\left(s_{2}\right)$, and $h\left(s_{3}\right)$.10.   **Initialize** the payoff matrix H
11.   **Update** the strategy proportion Y based on Equation (1)12.   **for** ($i=1:i<n_{1}+1$) **do**13.    **if** (t%2==0) **then**14.     **if** ($r_{3}<0.5$) **then**15.      **if** ($\left|A\right|<1$) **then**16.       **Update** position of current search agent as Equation (7)17.      **else**18.       **Select** three random search agents $X_{paddle1}$, $X_{paddle2}$, and $X_{paddle3}$
19.       **Update** (z) by the exponential form of Equation (9)20.       **Update** position of current search agent as Equation (8)21.      **end if**22.     **else**23.      **Update** position of current search agent as Equation (10)24.     **end if**25.    **else**26.     **Update** position of current search agent as Equation (13)27.    **end if**28.   **end for**29.   **for** ($i=1:i<n_{2}+1$) **do**30.    **if** (t%2==0) **then**31.     **Calculate**
$X_{1}$, $X_{2}$, and $X_{3}$ by the Equation (11)32.     **Update** individual position as Equation (12)33.    **else**34.     **Update** position of current search agent as Equation (13)35.    **end if**36.   **end for**37.   **Calculate** objective function $F_{n}$ for each agent $X_{i}$
38.   **Update** parameters39.   **Set**
t=t+1
40.   **Adjust** beyond the search space solutions41.   **if** (Best is same as previous two iterations) **then**42.    **Increase** solutions of exploration group ($n_{1}$)43.    **Decrease** solutions of exploitation group ($n_{2}$)44.   **end if**45. **end while**46. **Return** best agent P


## 3. Results

### 3.1. Comparison and Analysis of Test Functions

This section compares EGGO with seven classic algorithms (standard GGO [10], GWO [7], MFO [22], SSA [23], WOA [9], HHO [24], and PSO [3]) using benchmark functions and the CEC 2022 test suite. Simulations are run on an Intel(R) Core (TM) i7-14650HX processor with Windows 11, 32GB RAM, and a 2.2GHz processor speed. The algorithms are implemented in MATLAB R2024b, with parameters set as shown in Table 1.

#### 3.1.1. Benchmark Test Functions

Among the 23 benchmark test functions, we selected 8 (F2, F5, F8, F11, F14, F17, F20, and F23) according to an arithmetic sequence as the comparison test functions for this stage. The 3D images of the test functions and the convergence curves of the algorithms are shown in Figure 4.

From the results shown in Figure 4, it can be observed that EGGO significantly outperforms the other algorithms in unimodal (F2, F5) and multimodal functions (F8, F11). The advantages in mixed functions (F14, F17) and combined functions (F20, F23) are reduced. It is notable that in the 30-repetition experiments of each benchmark function group, the performance of SSA is not stable. For example, in Figure 4a, it can be observed that the algorithm stops converging when the number of iterations reaches 211. In each of the 30-repetition experiments of the benchmark test functions, the number of times EGGO ranks first is all over 28, demonstrating strong algorithm performance.

#### 3.1.2. CEC 2022 Test Suite

In this section, the above eight algorithms will be compared and tested using the CEC 2022 test suite. The function information of the test suite is shown in Table 2 below.

The test suite dimension is set to 10 dimensions (10D), and the experimental results are shown in Table 3. According to Table 2, in the 10D tests of 12 functions, EGGO achieves the following rankings:

Average Value (Ave.): first place in eight functions (F2, F3, F4, F6, F7, F9, F11, F12); second place in two functions (F8, F10); third place in one function (F5); sixth place in one function (F1).

Standard Deviation (Std.): first place in ten functions (F1, F3, F4, F5, F6, F7, F8, F9, F11, F12); second place in two functions (F2, F10).

Execution Time (Time): first place in two functions (F7, F11); fifth place in three functions (F8, F9, F12); sixth place in one function (F10); seventh place in three functions (F3, F4, F5); eighth place in three functions (F1, F2, F6).

Best Value (Best): first place in nine functions (F3, F4, F5, F6, F7, F8, F9, F11, F12); third place in two functions (F1, F2); seventh place in one function (F10).

Based on the rankings in Table 3, Figure 5 visualizes the performance of all six algorithms across four metrics: Ave., Std., Time, and Best.

The test suite dimension is set to 20 dimensions (20D), and the experimental results are shown in Table 4. According to Table 4, in the 20D tests of 12 functions, EGGO achieves the following rankings:

Average Value (Ave.): first place in eleven functions (F1, F2, F3, F4, F5, F6, F7, F8, F10, F11, F12); second place in one function (F9).

Standard Deviation (Std.): first place in seven functions (F2, F3, F6, F7, F8, F10, F12); second place in three functions (F1, F5, F9); third place in two functions (F4, F11).

Execution Time (Time): first place in four functions (F1, F3, F5, F8); second place in one function (F2); fifth place in three functions (F9, F10, F11); sixth place in two functions (F7, F12); seventh place in two functions (F4, F6).

Best Value (Best): first place in eight functions (F2, F4, F5, F6, F8, F9, F10, F12); second place in three functions (F3, F7, F11); third place in one function (F1).

Based on the rankings in Table 4, Figure 6 visualizes the performance of all six algorithms across four metrics: Ave., Std., Time, and Best.

Comprehensive evaluation of algorithm performance across the CEC 2022 benchmark suite indicates that EGGO significantly outperforms the other algorithms, particularly in terms of convergence speed and precision. When the dimensionality of the test suite increases from 10D to 20D, EGGO shows more remarkable advantages. Its rankings for mean value, variance, time consumption, and best score in the 20D tests are significantly better than those in the 10D tests. This highlights EGGO’s superior performance in handling high-dimensional problems compared to the other algorithms. The visualization results of this analysis are presented in Figure 7.

### 3.2. Engineering Applications of EGGO

To further evaluate the EGGO algorithm’s applicability to engineering design problems, this paper assesses its performance on three such problems: gear train design, tension–compression spring design, and three-bar truss design. In the experiments, the design variables of each problem serve as individual information in the algorithm, with the design model acting as the objective function for optimization. The results of the EGGO algorithm are compared with those of other algorithms to demonstrate its superiority in solving engineering problems. These algorithms originate from various sources in the literature and include PSO [3], GWO [7], SSA [22], WOA [9], MFO [22], HHO [24], GMO [25], KABC [26], ALO [27], GOA [28], CS [29], MBA [30], GSA [31], IAPSO [32], and DMMFO [33].

#### 3.2.1. Tension/Compression Spring Design Problem

As shown in Figure 8, the tension/compression spring design (TCSD) problem is a constrained optimization task aimed at minimizing spring volume under constant tension/compression loads, as described by Belegundu and Arora [34]. It involves three design variables: the number of coils (*L*), the coil diameter (*d*), and the wire diameter (*w*).

The calculation model of the tension/compression spring is as follows:(14)$$Minimize\;f_{1}\left(\omega ,d,L\right)=\left(L+2\right)\omega^{2}d$$

It is subject to the following constraints:(15)$$\left\{\begin{array}{c} g_{1}=1-\frac{d^{3}+L}{71.783\omega^{4}}\leq 0;\\ g_{2}=\frac{4d^{3}-\omega L}{12.566\left(dL^{3}-\omega^{4}\right)}\leq 0;\\ g_{3}=1-\frac{140.45\omega }{d^{2}L}\leq 0;\\ g_{4}=\frac{2\left(\omega +d\right)}{3}-1\leq 0. \end{array}\right.$$

The value ranges of the other three variables are as follows:0.05≤ω≤2.0,0.25≤d≤1.3,2.0≤L≤15.

For constrained optimization, EGGO employs a static penalty function method. The fitness function is reformulated as $F\left(X\right)=f\left(X\right)+\lambda \sum_{j=1}^{n}{\left[max\left(0,g_{j}\left(X\right)\right)\right]}^{2}$, where $f\left(X\right)$ is the primary objective; $g_{j}\left(X\right)$ denotes n design constraints; and the penalty coefficient $\lambda =10^{6}$ ensures strong infeasibility rejection. This mechanism transforms constrained problems into unoptimization via objective space mapping.

As indicated in Table 5, the computational outcomes of the EGGO algorithm for the tension/compression spring design problem are compared with those from seven other algorithms. All variables in the results satisfy the constraints. The EGGO algorithm demonstrates significant superiority in performance.

#### 3.2.2. Gear Train Design Problem

Gear train design (GTD) is a classic engineering design problem in mechanical transmission [35,36]. Its objective is to determine the number of teeth on each gear in the transmission system based on a reasonable gear ratio. The gear train structure is shown in Figure 9. The design variables for GTD consist of the number of teeth on four gears, denoted as *x*_1,_
*x*_2_, *x*_3_, and *x*_4_.

The specific mathematical model is as follows:(16)$$Consider\;\vec{x}=\left[x_{1},x_{2},x_{3},x_{4}\right]=\left[A,B,C,D\right]$$(17)$$Minimize\;f_{2}\left(x\right)={\left[\frac{1}{6.931}-\frac{x_{3}x_{2}}{x_{1}x_{3}}\right]}^{2}$$

It is subject to $12\leq x_{1},x_{2},x_{3},x_{4}\leq 60$.

The solution to the gear train design problem is unique and must be an integer. As shown in Table 6, EGGO yields the same optimal results as GMO, ALO, IAPSO, and MBA. This demonstrates that EGGO’s solution is both optimal and feasible.

#### 3.2.3. Three-Bar Truss Design Problem

TA schematic diagram of the three-bar truss design (T-BTD) is shown in Figure 10, and T-BTD is a mechanical optimization problem [26]. The objective of this problem is to minimize the weight of the three-bar truss structure while meeting the constraints of stress and loading force. The optimization variables of this problem are the cross-sectional areas of the connecting rods (*x*_1_, *x*_2_).

The specific optimization objective function is:(18)$$Minimize\;f_{3}\left(x\right)=\left(2\sqrt{2}x_{1}+x_{2}\right)*l$$

In the formula, *l* represents the distance between the connecting rods, and *l* = 100 cm, *x*_1_*, x*_2_ ∈ [0, 1].

During the T-BTD optimization process, the design variables are constrained from three aspects: structural stress, material deflection, and buckling. The three constraint formulas are as follows:(19)$$\left\{\begin{array}{c} g_{1}\left(x\right)=\frac{\sqrt{2}x_{1}+x_{2}}{\sqrt{2}x_{1}^{2}+2x_{1}x_{2}},P-\sigma \leq 0;\\ g_{2}\left(x\right)=\frac{x_{2}}{\sqrt{2}x_{1}^{2}+2x_{1}x_{2}},P-\sigma \leq 0;\\ g_{3}\left(x\right)=\frac{1}{\sqrt{2}x_{2}+x_{1}},P-\sigma \leq 0. \end{array}\right.$$

Here, *P* = 2 KN/cm^2^ and *σ* = 2 KN/cm^2^.

Based on Equations (18) and (19), the EGGO algorithm was applied to solve the three-bar truss design problem, with results presented in Table 7. When compared to the other algorithms, EGGO achieved the same optimal fitness as GMO, ALO, and GSA. Additionally, the solution values *x*_1_ and *x*_2_ satisfy the constraints. These experimental results demonstrate the EGGO algorithm’s capability to effectively solve the three-bar truss design problem.

## 4. Conclusions

In response to the shortcomings of the traditional GGO, such as being prone to getting trapped in local optima and having a slow convergence speed, an EGGO algorithm based on evolutionary game theory is proposed. EGGO significantly enhances the global search ability and convergence speed by introducing the dynamic strategy adjustment mechanism from evolutionary game theory. At the same time, it adopts strategies such as a dynamic grouping mechanism, random mutation, and local search enhancement to further improve the efficiency and robustness of the algorithm. The dynamic grouping mechanism of EGGO dynamically adjusts the number of the exploration group and development group according to the fitness distribution of the population to balance global searching and local development; random mutation or re-initialization of some individuals maintains the diversity of the population and prevents premature convergence; and the introduction of a local search ability improves the local development ability of the algorithm.

The EGGO algorithm was compared with seven classic algorithms: standard GGO, GWO, MFO, SSA, WOA, HHO, and PSO. The results indicate that EGGO surpasses the others in multiple standard test functions and the CEC 2022 benchmark suite. It shows excellent performance in convergence accuracy and speed. EGGO is significantly better in unimodal and multimodal functions. Though its advantage is slightly less in hybrid and composite functions, it still ranks first in over 28 of 30 runs per group, proving its strong performance. In the CEC 2022 tests for 10D and 20D, EGGO achieved excellent results. Among the 12 functions in the 10D test, EGGO ranked first in mean value, standard deviation, execution time, and best value. In the 20D test, it also performed outstandingly in these metrics. This indicates that EGGO outperforms other algorithms in handling high-dimensional problems.

In engineering applications, EGGO was applied to the engineering design problems of tension/compression spring design, gear train design, and three-bar truss design. The experimental results indicate that EGGO can efficiently solve these problems while satisfying constraints, with results either comparable to or better than other advanced algorithms. This demonstrates the feasibility and effectiveness of EGGO in practical engineering optimization.

In this paper, the following contributions are proposed:

1. An EGGO algorithm is proposed. By incorporating dynamic strategy adjustment from evolutionary game theory, it improves the algorithm’s adaptability and strategy selection ability. This significantly boosts global search efficiency and convergence speed.

2. EGGO uses a dynamic grouping mechanism. Based on population fitness distribution, it adjusts exploration and exploitation groups to balance global and local searches. This avoids premature convergence and ensures excellent performance in high-dimensional problems.

3. Random mutation and local search enhancement strategies are adopted. These maintain population diversity, prevent early convergence, and improve local exploitation ability.

Despite the remarkable performance improvements achieved by EGGO, there remains scope for further enhancement. Future work will focus on fine-tuning the algorithm’s parameters to accommodate diverse optimization scenarios and integrating additional adaptive strategies to bolster the algorithm’s robustness and applicability. Furthermore, the potential of EGGO will be explored in a broader range of practical applications, such as engineering design, resource allocation, and scheduling problems. In summary, the EGGO algorithm proposed in this paper has significantly advanced the development of GGO and laid a solid foundation for future innovation in the field of computational intelligence and optimization.

## Figures and Tables

**Figure 1 biomimetics-10-00545-f001:**
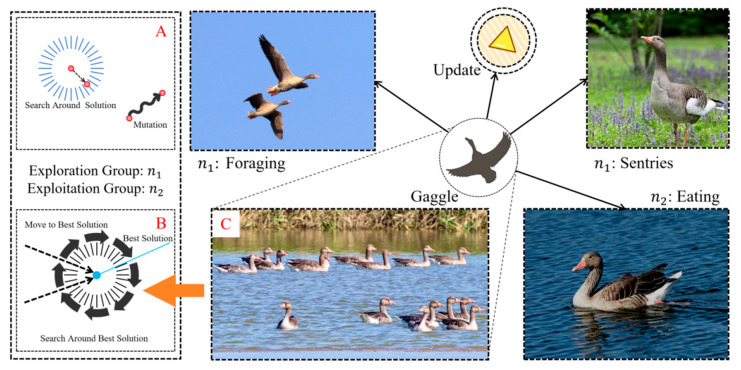
Greylag Goose Optimization exploration, exploitation, and dynamic groups. (**A**) Exploration group; (**B**) Exploitation group; (**C**) Dynamic group in real life.

**Figure 2 biomimetics-10-00545-f002:**
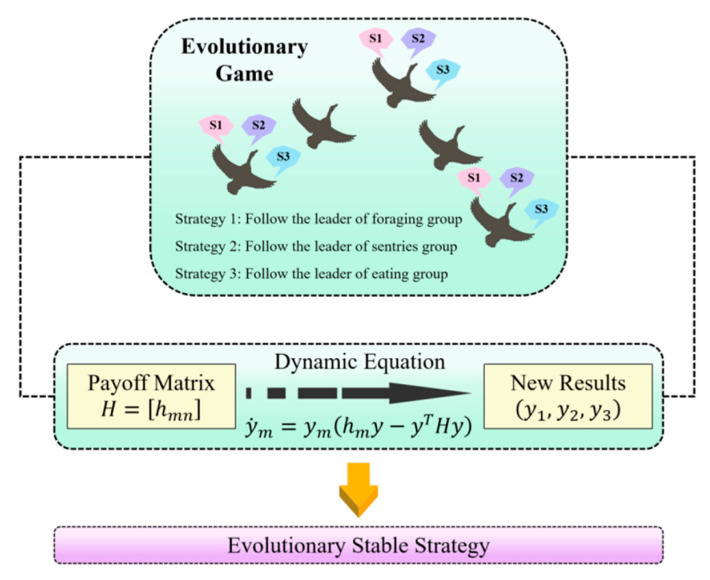
Schematic diagram of EGGO in detail.

**Figure 3 biomimetics-10-00545-f003:**
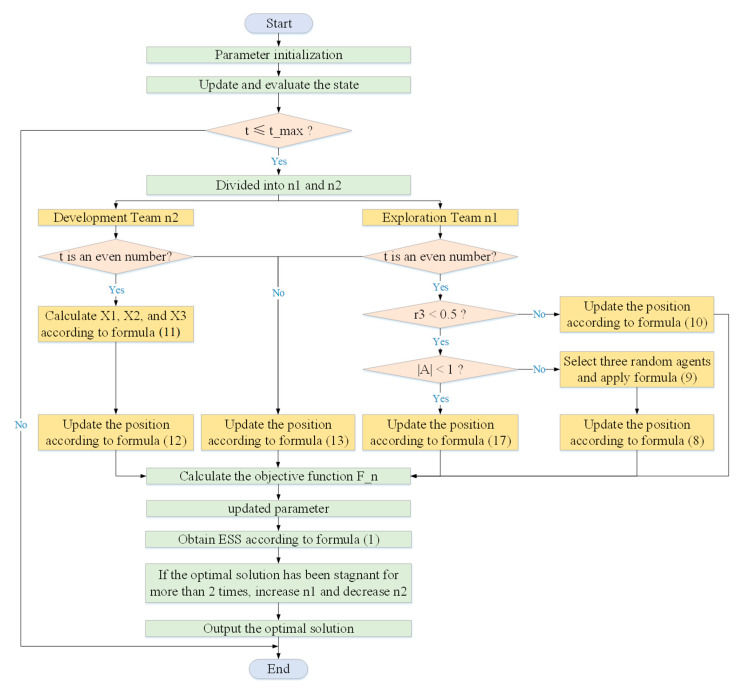
Flowchart of EGGO.

**Figure 4 biomimetics-10-00545-f004:**
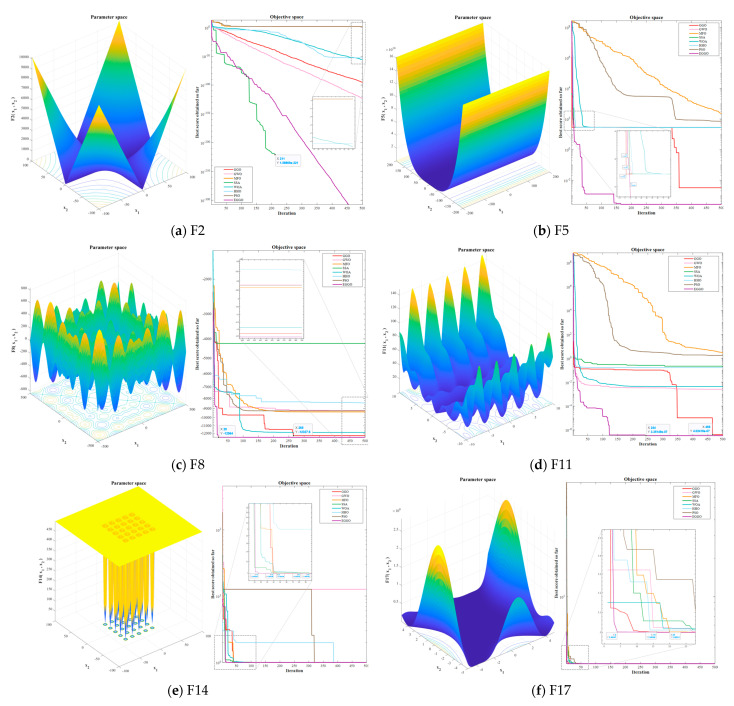
Test results in benchmark test function.

**Figure 5 biomimetics-10-00545-f005:**
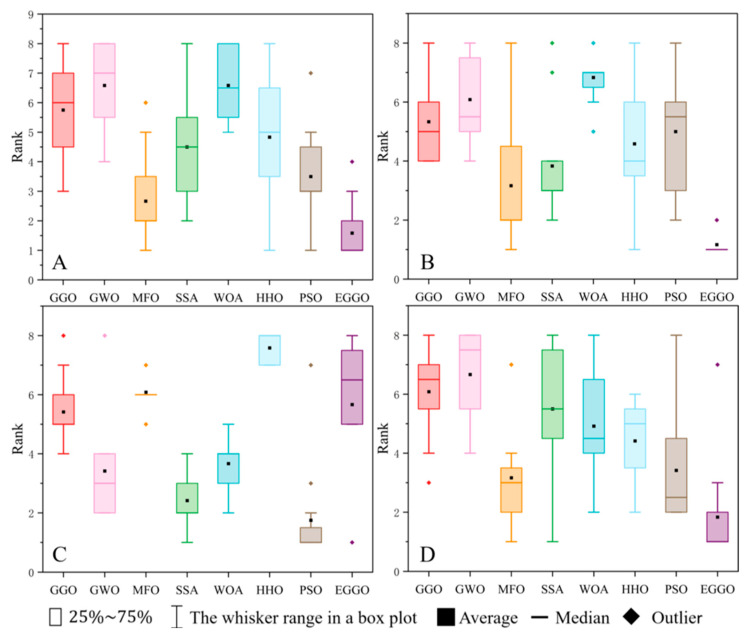
Test result (CEC 2022 10D) ranking: (**A**) Ave.; (**B**) Std.; (**C**) Time; (**D**) Best.

**Figure 6 biomimetics-10-00545-f006:**
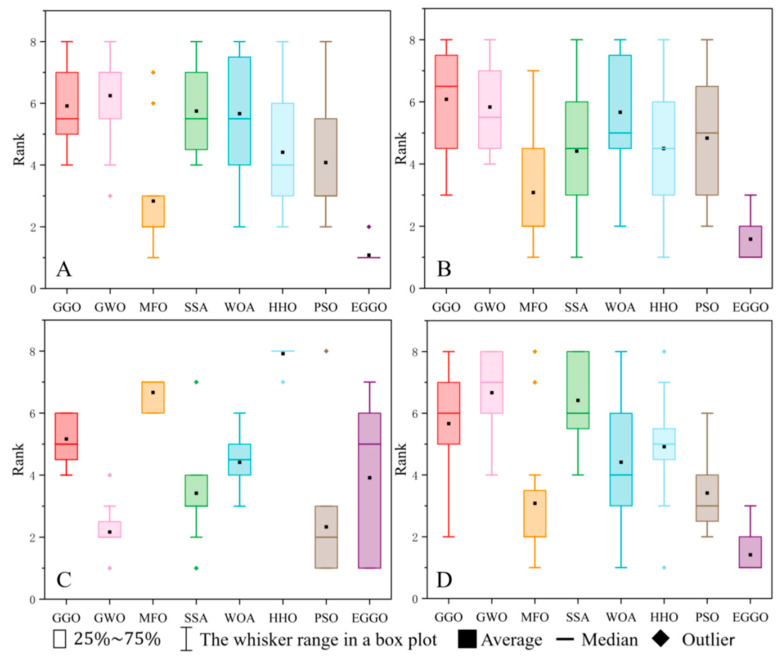
Test result (CEC 2022 20D) ranking: (**A**) Ave.; (**B**) Std.; (**C**) Time; (**D**) Best.

**Figure 7 biomimetics-10-00545-f007:**
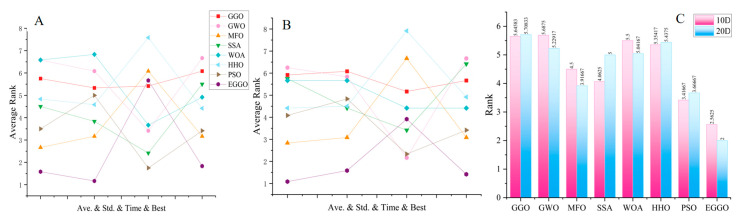
Test result: (**A**) average rank in 10D test; (**B**) average rank in 20D test; (**C**) total average rank in 10D and 20D test.

**Figure 8 biomimetics-10-00545-f008:**
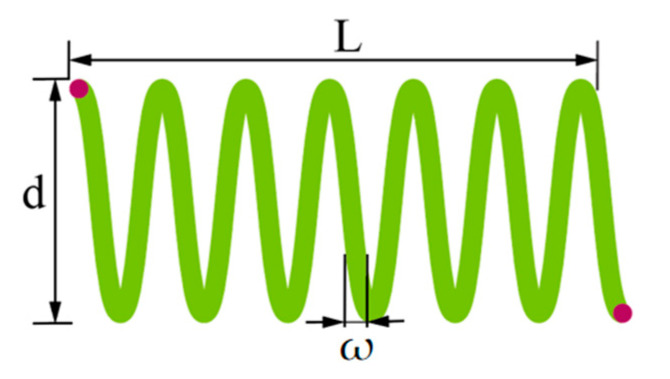
Schematic of the tension/compression spring.

**Figure 9 biomimetics-10-00545-f009:**
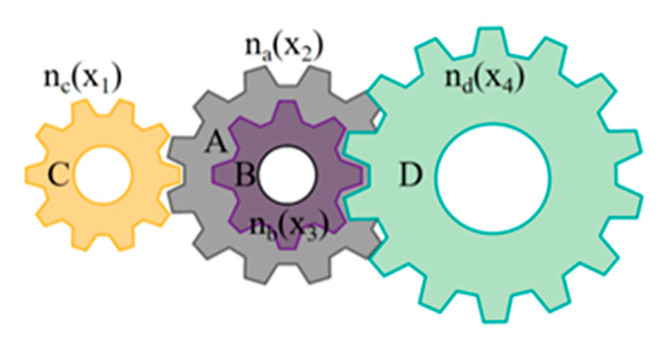
Transmission diagram of the gear train.

**Figure 10 biomimetics-10-00545-f010:**
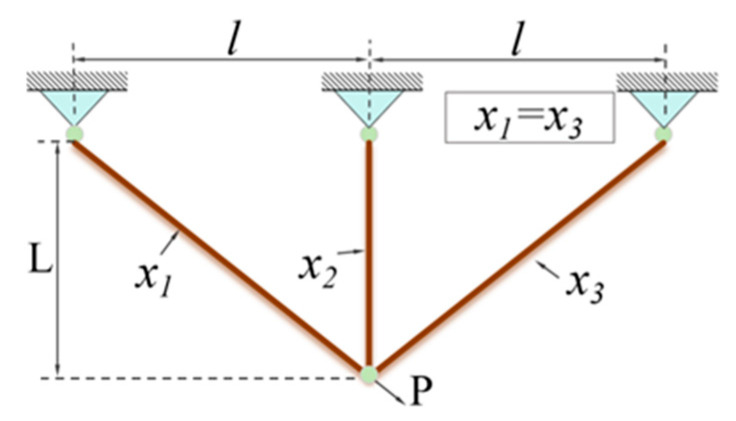
Schematic of three-bar truss mechanism.

**Table 1 biomimetics-10-00545-t001:** Settings of algorithm parameters under 500 iterations and a population size of 100.

Algorithm	Parameter(s)	Value(s)
GGO	$r_{1},\;r_{2},\;r_{3},\;r_{4},\;r_{5}$	$\left[0,\;1\right]$
	$\omega_{1},\;\omega_{2},\;\omega_{3},\;\omega_{4}$	$\left[0,\;2\right]$
GWO	a	2 to 0
MFO	a	−1 to −2
SSA	PD	0.7
	SD	0.2
	ST	0.8
	CD	0.3
HHO	$E_{1}$	2 to 0
	$E_{0}$	$\left[-1,\;1\right]$
PSO	$W_{max},\;W_{min}$	0.9, 0.6
	$C_{1},\;C_{2}$	2, 2
EGGO	$r_{1},\;r_{2},\;r_{3},\;r_{4},\;r_{5}$	$\left[0,\;1\right]$
	$\omega_{1},\;\omega_{2},\;\omega_{3},\;\omega_{4}$	$\left[0,\;2\right]$

**Table 2 biomimetics-10-00545-t002:** Details of the CEC 2022.

Type	No.	Functions	Dimension	$F_{min}$
Unimodal	1	Shifted and fully rotated Zakharov’s function	10 and 20	300
Multimodal	2	Shifted and fully rotated Rosenbrock’s function	10 and 20	400
	3	Shifted and fully rotated expanded Schaffer’s F6 function	10 and 20	600
	4	Shifted and fully rotated non-continuous Rastrigin’s function	10 and 20	800
	5	Shifted and fully rotated Levy’s function	10 and 20	900
Hybrid	6	Hybrid function 1 (N = 3)	10 and 20	1800
	7	Hybrid function 2 (N = 6)	10 and 20	2000
	8	Hybrid function 3 (N = 5)	10 and 20	2200
Composition	9	Composition function 1 (N = 5)	10 and 20	2300
	10	Composition function 2 (N = 4)	10 and 20	2400
	11	Composition function 3 (N = 5)	10 and 20	2600
	12	Composition function 4 (N = 6)	10 and 20	2700

**Table 3 biomimetics-10-00545-t003:** Results of the CEC 2022 (10D).

Function	GGO				GWO			
	Ave.	Std.	Time	Best	Ave.	Std.	Time	Best
F1	8.9718 × 10^3^	1.5981 × 10^4^	5.5247 × 10^−3^	2.7181 × 10^3^	1.3111 × 10^4^	4.2103 × 10^3^	4.5857 × 10^−3^	5.2255 × 10^3^
F2	5.7769 × 10^2^	1.0435 × 10^2^	5.0988 × 10^−3^	4.3759 × 10^2^	6.7885 × 10^2^	1.8679 × 10^2^	4.6382 × 10^−3^	4.4534 × 10^2^
F3	6.4261 × 10^2^	1.1480 × 10^1^	1.1234 × 10^−2^	6.2053 × 10^2^	6.3172 × 10^2^	9.4214 × 10^0^	1.0332 × 10^−2^	6.1625 × 10^2^
F4	8.4191 × 10^2^	1.0476 × 10^1^	7.5605 × 10^−3^	8.2221 × 10^2^	8.4370 × 10^2^	1.1005 × 10^1^	6.5427 × 10^−3^	8.2070 × 10^2^
F5	1.4666 × 10^3^	2.1263 × 10^2^	7.8058 × 10^−3^	1.0897 × 10^3^	1.2911 × 10^3^	2.3521 × 10^2^	6.6609 × 10^−3^	9.5376 × 10^2^
F6	8.8523 × 10^5^	1.7495 × 10^6^	5.9593 × 10^−3^	3.3302 × 10^3^	3.1509 × 10^6^	7.1051 × 10^6^	5.6340 × 10^−3^	2.9283 × 10^3^
F7	2.0917 × 10^3^	2.8859 × 10^1^	6.0866 × 10^−2^	2.0438 × 10^3^	2.0877 × 10^3^	3.2519 × 10^1^	1.1995 × 10^−2^	2.0457 × 10^3^
F8	2.2397 × 10^3^	1.2090 × 10^1^	2.0055 × 10^−2^	2.2204 × 10^3^	2.2390 × 10^3^	2.3285 × 10^1^	1.5010 × 10^−2^	2.2247 × 10^3^
F9	2.6888 × 10^3^	4.8706 × 10^1^	1.3135 × 10^−2^	2.5514 × 10^3^	2.7271 × 10^3^	5.9966 × 10^1^	1.1441 × 10^−2^	2.5818 × 10^3^
F10	2.5896 × 10^3^	8.5976 × 10^1^	1.2604 × 10^−2^	2.5007 × 10^3^	2.7194 × 10^3^	3.6536 × 10^2^	1.1074 × 10^−2^	2.5037 × 10^3^
F11	3.2998 × 10^3^	4.1118 × 10^2^	2.0037 × 10^−2^	2.8391 × 10^3^	4.0570 × 10^3^	3.8145 × 10^2^	1.6217 × 10^−2^	3.1253 × 10^3^
F12	2.8958 × 10^3^	3.6440 × 10^1^	6.6573 × 10^−2^	2.8694 × 10^3^	2.9202 × 10^3^	3.4289 × 10^1^	1.6469 × 10^−2^	2.8729 × 10^3^
Function	MFO				SSA			
	Ave.	Std.	Time	Best	Ave.	Std.	Time	Best
F1	1.1746 × 10^4^	6.4984 × 10^3^	7.9990 × 10^−3^	2.2914 × 10^3^	6.8049 × 10^3^	2.3104 × 10^3^	4.7821 × 10^−3^	1.0998 × 10^3^
F2	4.1524 × 10^2^	1.7718 × 10^1^	7.9137 × 10^−3^	4.0764 × 10^2^	5.0932 × 10^2^	6.4575 × 10^1^	4.7256 × 10^−3^	4.3617 × 10^2^
F3	6.0385 × 10^2^	4.1744 × 10^0^	1.3584 × 10^−2^	6.0074 × 10^2^	6.3476 × 10^2^	8.7952 × 10^0^	9.9959 × 10^−3^	6.2033 × 10^2^
F4	8.3243 × 10^2^	1.4684 × 10^1^	9.8440 × 10^−3^	8.2561 × 10^2^	8.4346 × 10^2^	9.8422 × 10^0^	6.5236 × 10^−3^	8.3317 × 10^2^
F5	9.9985 × 10^2^	1.3416 × 10^2^	1.0137 × 10^−2^	9.0080 × 10^2^	1.3829 × 10^3^	2.4189 × 10^2^	6.7142 × 10^−3^	1.0154 × 10^3^
F6	4.4705 × 10^3^	2.0481 × 10^3^	1.0395 × 10^−2^	1.9312 × 10^3^	9.6276 × 10^6^	7.6058 × 10^6^	5.8453 × 10^−3^	2.0961 × 10^5^
F7	2.0280 × 10^3^	1.0856 × 10^1^	1.5446 × 10^−2^	2.0212 × 10^3^	2.0812 × 10^3^	1.8711 × 10^1^	1.1743 × 10^−2^	2.0442 × 10^3^
F8	2.2253 × 10^3^	5.0467 × 10^0^	1.7968 × 10^−2^	2.2044 × 10^3^	2.2432 × 10^3^	7.5604 × 10^0^	1.4752 × 10^−2^	2.2302 × 10^3^
F9	2.5359 × 10^3^	1.9028 × 10^1^	1.5121 × 10^−2^	2.5293 × 10^3^	2.6328 × 10^3^	4.2446 × 10^1^	1.1410 × 10^−2^	2.5370 × 10^3^
F10	2.5092 × 10^3^	3.1289 × 10^1^	1.4872 × 10^−2^	2.5004 × 10^3^	2.5710 × 10^3^	7.7876 × 10^1^	1.0658 × 10^−2^	2.5016 × 10^3^
F11	3.2059 × 10^3^	3.0766 × 10^2^	1.9802 × 10^−2^	2.7544 × 10^3^	3.0473 × 10^3^	2.8749 × 10^2^	1.5646 × 10^−2^	2.7994 × 10^3^
F12	2.9037 × 10^3^	3.8110 × 10^1^	2.9144 × 10^−2^	2.8715 × 10^3^	2.8801 × 10^3^	1.6583 × 10^1^	1.6196 × 10^−2^	2.8676 × 10^3^
Function	WOA				HHO			
	Ave.	Std.	Time	Best	Ave.	Std.	Time	Best
F1	3.3560 × 10^4^	1.4314 × 10^4^	5.1796 × 10^−3^	6.4874 × 10^3^	6.2245 × 10^3^	1.4228 × 10^3^	1.4660 × 10^−2^	2.1536 × 10^3^
F2	5.2604 × 10^2^	1.1783 × 10^2^	5.1717 × 10^−3^	4.1115 × 10^2^	5.3526 × 10^2^	1.0068 × 10^2^	1.3786 × 10^−2^	4.1830 × 10^2^
F3	6.4022 × 10^2^	1.3011 × 10^1^	1.0706 × 10^−2^	6.1425 × 10^2^	6.4120 × 10^2^	1.1798 × 10^1^	2.7284 × 10^−2^	6.1749 × 10^2^
F4	8.4883 × 10^2^	1.3810 × 10^1^	6.9401 × 10^−3^	8.2021 × 10^2^	8.2818 × 10^2^	8.4948E+00	1.9141 × 10^−2^	8.1322 × 10^2^
F5	1.5849 × 10^3^	3.5923 × 10^2^	7.1641 × 10^−3^	1.0663 × 10^3^	1.4879 × 10^3^	1.9571 × 10^2^	1.9885 × 10^−2^	1.0407 × 10^3^
F6	7.5709 × 10^4^	3.0266 × 10^5^	5.7107 × 10^−3^	2.5663 × 10^3^	1.6960 × 10^4^	1.5749 × 10^4^	1.6687 × 10^−2^	2.4769 × 10^3^
F7	2.0915 × 10^3^	3.3568 × 10^1^	1.2637 × 10^−2^	2.0425 × 10^3^	2.0799 × 10^3^	3.7449 × 10^1^	3.2018 × 10^−2^	2.0288 × 10^3^
F8	2.2453 × 10^3^	2.2034 × 10^1^	1.5100 × 10^−2^	2.2271 × 10^3^	2.2365 × 10^3^	1.1173 × 10^1^	3.8780 × 10^−2^	2.2220 × 10^3^
F9	2.6465 × 10^3^	5.0501 × 10^1^	1.1429 × 10^−2^	2.5445 × 10^3^	2.6576 × 10^3^	4.7668 × 10^1^	2.8653 × 10^−2^	2.5412 × 10^3^
F10	2.6658 × 10^3^	3.0651 × 10^2^	1.1056 × 10^−2^	2.5004 × 10^3^	2.6260 × 10^3^	1.7383 × 10^2^	2.7131 × 10^−2^	2.5010 × 10^3^
F11	3.3174 × 10^3^	4.1017 × 10^2^	1.6037 × 10^−2^	2.7653 × 10^3^	3.1280 × 10^3^	3.2571 × 10^2^	3.6550 × 10^−2^	2.7317 × 10^3^
F12	2.9193 × 10^3^	4.3466 × 10^1^	1.6161 × 10^−2^	2.8682 × 10^3^	2.9453 × 10^3^	6.4707 × 10^1^	4.0554 × 10^−2^	2.8705 × 10^3^
Function	PSO				EGGO			
	Ave.	Std.	Time	Best	Ave.	Std.	Time	Best
F1	9.7990 × 10^3^	5.9490 × 10^3^	4.5264 × 10^−3^	2.6880 × 10^3^	9.0246 × 10^3^	2.1287 × 10^3^	1.5709 × 10^−2^	2.1574 × 10^3^
F2	4.4942 × 10^2^	6.2600 × 10^1^	4.5352 × 10^−3^	4.0791 × 10^2^	4.1372 × 10^2^	1.5217 × 10^1^	1.6228 × 10^−2^	4.0991 × 10^2^
F3	6.0719 × 10^2^	5.6420 × 10^0^	1.0123 × 10^−2^	6.3648 × 10^2^	6.0295 × 10^2^	3.8314 × 10^0^	2.1082 × 10^−2^	6.0067 × 10^2^
F4	8.3708 × 10^2^	1.2480 × 10^1^	6.2257 × 10^−3^	8.1230 × 10^2^	8.2084 × 10^2^	7.9549 × 10^0^	1.7677 × 10^−2^	8.0714 × 10^2^
F5	9.9364 × 10^2^	2.0986 × 10^2^	6.4482 × 10^−3^	9.0027 × 10^2^	1.1621 × 10^3^	1.1416 × 10^2^	1.7785 × 10^−2^	9.0007 × 10^2^
F6	6.1313 × 10^3^	2.2224 × 10^3^	5.1812 × 10^−3^	1.8941 × 10^3^	4.4205 × 10^3^	1.8767 × 10^3^	1.7495 × 10^−2^	1.8541 × 10^3^
F7	2.0381 × 10^3^	3.3524 × 10^1^	1.1919 × 10^−2^	2.0213 × 10^3^	2.0265 × 10^3^	8.6546 × 10^0^	1.1643 × 10^−2^	2.0182 × 10^3^
F8	2.2308 × 10^3^	1.8176 × 10^1^	1.4598 × 10^−2^	2.2068 × 10^3^	2.2253 × 10^3^	4.0303 × 10^0^	1.6902 × 10^−2^	2.2043 × 10^3^
F9	2.5650 × 10^3^	6.2866 × 10^1^	1.0617 × 10^−2^	2.5293 × 10^3^	2.5227 × 10^3^	1.1393 × 10^1^	1.1867 × 10^−2^	2.5172 × 10^3^
F10	2.6067 × 10^3^	1.8315 × 10^2^	1.0071 × 10^−2^	2.5008 × 10^3^	2.5336 × 10^3^	5.1820 × 10^1^	1.2894 × 10^−2^	2.5021 × 10^3^
F11	3.4855 × 10^3^	4.5981 × 10^2^	3.0300 × 10^−2^	2.8209 × 10^3^	2.9557 × 10^3^	2.7451 × 10^2^	1.5464 × 10^−2^	2.6972 × 10^3^
F12	2.8724 × 10^3^	1.2959 × 10^1^	1.5298 × 10^−2^	2.8624 × 10^3^	2.8634 × 10^3^	1.3006 × 10^0^	1.9950 × 10^−2^	2.8597 × 10^3^

**Table 4 biomimetics-10-00545-t004:** Results of the CEC 2022 (20D).

Function	GGO				GWO			
	Ave.	Std.	Time	Best	Ave	std	Time	Best
F1	5.9176 × 10^4^	3.3688 × 10^4^	6.4054 × 10^−3^	1.7561 × 10^4^	5.2474 × 10^4^	1.7347 × 10^4^	5.8229 × 10^−3^	2.1648 × 10^4^
F2	1.6682 × 10^3^	5.1504 × 10^2^	6.2956 × 10^−3^	8.6866 × 10^2^	1.6142 × 10^3^	6.6266 × 10^2^	5.5554 × 10^−3^	8.3519 × 10^2^
F3	6.7649 × 10^2^	1.1363 × 10^1^	1.7646 × 10^−2^	6.4770 × 10^2^	6.6671 × 10^2^	1.2263 × 10^1^	1.6815 × 10^−2^	6.3532 × 10^2^
F4	9.5411 × 10^2^	1.9946 × 10^1^	1.0233 × 10^−2^	9.1132 × 10^2^	9.5686 × 10^2^	1.9856 × 10^1^	9.2864 × 10^−3^	9.1853 × 10^2^
F5	3.4490 × 10^3^	4.3306 × 10^2^	1.0047 × 10^−2^	2.2112 × 10^3^	3.3415 × 10^3^	5.7951 × 10^2^	9.2455 × 10^−3^	1.8887 × 10^3^
F6	6.4271 × 10^8^	5.5643 × 10^8^	7.0105 × 10^−3^	2.0092E+07	6.5769 × 10^8^	6.1897 × 10^8^	6.2666 × 10^−3^	4.9632E+07
F7	2.2067 × 10^3^	5.7969 × 10^1^	2.0988 × 10^−2^	2.1064 × 10^3^	2.2292 × 10^3^	5.8759 × 10^1^	1.9661 × 10^−2^	2.1208 × 10^3^
F8	2.3474 × 10^3^	1.2507 × 10^2^	2.4538 × 10^−2^	2.2328 × 10^3^	2.3625 × 10^3^	1.1567 × 10^2^	2.2600 × 10^−2^	2.2329 × 10^3^
F9	2.9256 × 10^3^	1.6498 × 10^2^	2.3938 × 10^−2^	2.6696 × 10^3^	2.8730 × 10^3^	1.2788 × 10^2^	2.1279 × 10^−2^	2.6940 × 10^3^
F10	5.4755 × 10^3^	1.8551 × 10^3^	1.8726 × 10^−2^	2.5362 × 10^3^	6.2215 × 10^3^	1.2073 × 10^3^	1.6678 × 10^−2^	2.6349 × 10^3^
F11	7.8689 × 10^3^	8.6044 × 10^2^	3.2313 × 10^−2^	5.6132 × 10^3^	8.7527 × 10^3^	7.8209 × 10^2^	2.7213 × 10^−2^	6.6834 × 10^3^
F12	3.1950 × 10^3^	1.4512 × 10^2^	3.4304 × 10^−2^	2.9730 × 10^3^	3.3076 × 10^3^	1.8595 × 10^2^	3.0777 × 10^−2^	3.0239 × 10^3^
Function	MFO				SSA			
	Ave.	Std.	Time	Best	Ave	std	Time	Best
F1	6.1231 × 10^4^	1.1673 × 10^4^	1.2524 × 10^−2^	3.3050 × 10^4^	7.9260 × 10^4^	3.1556 × 10^4^	1.7645 × 10^−2^	3.5907 × 10^4^
F2	5.4734 × 10^2^	6.0904 × 10^1^	1.1413 × 10^−2^	4.6990 × 10^2^	1.0300 × 10^3^	2.1787 × 10^2^	5.7990 × 10^−3^	5.9555 × 10^2^
F3	6.2558 × 10^2^	7.3784 × 10^0^	2.2844 × 10^−2^	6.1151 × 10^2^	6.6635 × 10^2^	1.1044 × 10^1^	1.6953E × 10^−2^	6.3920 × 10^2^
F4	9.0440 × 10^2^	2.1010 × 10^1^	1.5567 × 10^−2^	8.6080 × 10^2^	9.6325 × 10^2^	1.4271 × 10^1^	9.3094 × 10^−3^	9.3560 × 10^2^
F5	3.0670 × 10^3^	1.0176 × 10^3^	1.5547 × 10^−2^	1.5266 × 10^3^	3.6631 × 10^3^	4.6332 × 10^2^	9.6323 × 10^−3^	2.2913 × 10^3^
F6	8.9806 × 10^6^	3.2434 × 10^7^	1.2335 × 10^−2^	3.7945 × 10^4^	1.4716 × 10^8^	1.0144 × 10^8^	6.6011 × 10^−3^	2.7310 × 10^7^
F7	2.1225 × 10^3^	5.0527 × 10^1^	2.6023 × 10^−2^	2.0362 × 10^3^	2.2096 × 10^3^	6.4204 × 10^1^	2.0051 × 10^−2^	2.0982 × 10^3^
F8	2.2556 × 10^3^	3.7816 × 10^1^	2.9029 × 10^−2^	2.2275 × 10^3^	2.3669 × 10^3^	7.0686 × 10^1^	2.2952 × 10^−2^	2.2473 × 10^3^
F9	2.4995 × 10^3^	1.5796 × 10^1^	2.7036 × 10^−2^	2.4817 × 10^3^	2.6849 × 10^3^	5.8066 × 10^1^	2.1603 × 10^−2^	2.5745 × 10^3^
F10	3.6510 × 10^3^	1.1102 × 10^3^	2.2880 × 10^−2^	2.5030 × 10^3^	5.3046 × 10^3^	2.0359 × 10^3^	1.7008 × 10^−2^	2.5315 × 10^3^
F11	1.6579 × 10^4^	8.2847 × 10^3^	3.3437 × 10^−2^	8.6969 × 10^3^	6.9695 × 10^3^	4.7637 × 10^2^	2.7133 × 10^−2^	5.6653 × 10^3^
F12	3.0669 × 10^3^	1.3933 × 10^1^	4.4198 × 10^−2^	2.9818 × 10^3^	3.1248 × 10^3^	9.3803 × 10^1^	3.0523 × 10^−2^	2.9939 × 10^3^
Function	WOA				HHO			
	Ave.	Std.	Time	Best	Ave	std	Time	Best
F1	5.4916 × 10^4^	1.8629 × 10^4^	6.4459 × 10^−3^	1.6798 × 10^4^	5.0829 × 10^4^	1.5932 × 10^4^	1.8694 × 10^−2^	2.1948 × 10^4^
F2	8.5476 × 10^2^	1.6965 × 10^2^	6.3828 × 10^−3^	5.8846 × 10^2^	8.7254 × 10^2^	1.5400 × 10^2^	1.6184 × 10^−2^	6.3827 × 10^2^
F3	6.7819 × 10^2^	1.2832 × 10^1^	1.7020 × 10^−2^	6.5280 × 10^2^	6.6708 × 10^2^	1.0961 × 10^1^	4.4547 × 10^−2^	6.3088 × 10^2^
F4	9.5966 × 10^2^	2.6966 × 10^1^	9.8217 × 10^−3^	8.9240 × 10^2^	8.9961 × 10^2^	1.4888 × 10^1^	2.5623 × 10^−2^	8.6119 × 10^2^
F5	4.5437 × 10^3^	1.3793 × 10^3^	1.0117 × 10^−2^	2.2502 × 10^3^	3.1558 × 10^3^	3.7781 × 10^2^	2.8117 × 10^−2^	2.1695 × 10^3^
F6	7.1896 × 10^7^	8.2773 × 10^7^	7.3713 × 10^−3^	3.3799 × 10^5^	1.7839 × 10^7^	3.7642 × 10^7^	1.9376 × 10^−2^	3.5391 × 10^5^
F7	2.2575 × 10^3^	8.9587 × 10^1^	2.0146 × 10^−2^	2.1066 × 10^3^	2.2271 × 10^3^	7.2493 × 10^1^	5.3109 × 10^−2^	2.1327 × 10^3^
F8	2.3488 × 10^3^	1.2004 × 10^2^	2.3399 × 10^−2^	2.2304 × 10^3^	2.3335 × 10^3^	1.0893 × 10^2^	5.8203 × 10^−2^	2.2318 × 10^3^
F9	2.6565 × 10^3^	6.8495 × 10^1^	2.1379 × 10^−2^	2.5218 × 10^3^	2.7010 × 10^3^	7.2822 × 10^1^	5.0356 × 10^−2^	2.5664 × 10^3^
F10	5.5397 × 10^3^	1.3492 × 10^3^	1.7146 × 10^−2^	2.5106 × 10^3^	4.8006 × 10^3^	1.7619 × 10^3^	4.3705 × 10^−2^	2.5284 × 10^3^
F11	5.6536 × 10^3^	5.8157 × 10^2^	2.7628 × 10^−2^	4.3616 × 10^3^	6.2412 × 10^3^	9.0108 × 10^2^	6.2003 × 10^−2^	4.1782 × 10^3^
F12	3.1741 × 10^3^	1.3144 × 10^2^	3.0692 × 10^−2^	3.0187 × 10^3^	3.3290 × 10^3^	2.0010 × 10^2^	7.5187 × 10^−2^	3.0188 × 10^3^
Function	PSO				EGGO			
	Ave.	Std.	Time	Best	Ave	std	Time	Best
F1	6.8451 × 10^4^	2.0562 × 10^4^	5.8537 × 10^−3^	2.9053 × 10^4^	3.7418 × 10^4^	1.1689 × 10^4^	5.8202 × 10^−3^	1.7921 × 10^4^
F2	7.1217 × 10^2^	2.2996 × 10^2^	1.6870 × 10^−2^	4.4896 × 10^2^	5.1170 × 10^2^	4.5991 × 10^1^	5.7311 × 10^−3^	4.3560 × 10^2^
F3	6.2776 × 10^2^	8.8523 × 10^0^	1.6637 × 10^−2^	6.1328 × 10^2^	6.2367 × 10^2^	6.7724 × 10^0^	1.5967 × 10^−2^	6.1260 × 10^2^
F4	9.4195 × 10^2^	2.6359 × 10^1^	9.0505 × 10^−3^	8.9326 × 10^2^	8.7182 × 10^2^	1.7680 × 10^1^	1.6379 × 10^−2^	8.4077 × 10^2^
F5	3.7861 × 10^3^	1.5398 × 10^3^	9.2301 × 10^−3^	1.6896 × 10^3^	2.7219 × 10^3^	3.9913 × 10^2^	9.1869 × 10^−3^	1.5199 × 10^3^
F6	9.6455 × 10^6^	1.6462 × 10^7^	6.5599 × 10^−3^	4.0381 × 10^3^	1.3426 × 10^6^	8.2665 × 10^5^	1.7603 × 10^−2^	2.1938 × 10^3^
F7	2.1349 × 10^3^	4.5206 × 10^1^	1.9406 × 10^−2^	2.0655 × 10^3^	2.1184 × 10^3^	3.6514 × 10^1^	2.1295 × 10^−2^	2.0420 × 10^3^
F8	2.3038 × 10^3^	7.3715 × 10^1^	2.2781 × 10^−2^	2.2305 × 10^3^	2.2486 × 10^3^	2.4609 × 10^1^	1.9803 × 10^−2^	2.2183 × 10^3^
F9	2.5862 × 10^3^	9.3564 × 10^1^	2.0743 × 10^−2^	2.4819 × 10^3^	2.5060 × 10^3^	5.1559 × 10^1^	2.3087 × 10^−2^	2.4649 × 10^3^
F10	4.3177 × 10^3^	1.1727 × 10^3^	1.6412 × 10^−2^	2.5212 × 10^3^	3.5931 × 10^3^	1.0167 × 10^3^	1.7544 × 10^−2^	2.5028 × 10^3^
F11	1.8980 × 10^4^	1.9458 × 10^4^	2.7436 × 10^−2^	4.4370 × 10^3^	5.5911 × 10^3^	5.8405 × 10^2^	3.1106 × 10^−2^	4.2239 × 10^3^
F12	3.0249 × 10^3^	4.5139 × 10^1^	2.9727 × 10^−2^	2.9527 × 10^3^	2.9533 × 10^3^	7.9448 × 10^0^	3.7432 × 10^−2^	2.9411 × 10^3^

**Table 5 biomimetics-10-00545-t005:** Comparison of the best solution for tension/compression spring design problem.

Algorithm	Optimal Values for Variables							f
	w	d	L	g_1_	g_2_	g_3_	g_4_	
GGO	0.05178	0.35885	11.171	−3.52 × 10^−4^	−1.33 × 10^−4^	−4.0555	−0.7262	0.0126724
PSO	0.0527	0.3809	10.03011	−0.0011	−0.0013	−4.0863	−0.7109	0.0127263
GWO	0.05173	0.35749	11.2594	−6.99 × 10^−4^	−4.80 × 10^−4^	−4.0492	−0.7272	0.0126845
SSA	0.0507	0.3319	12.98	−5.31 × 10^−4^	−0.0035	−3.9801	−0.7449	0.0127801
WOA	0.05173	0.35764	11.24	−2.32 × 10^−4^	−1.43 × 10^−4^	−4.0537	−0.7271	0.0126712
MFO	0.05172	0.35746	11.31	−0.0057	−5.64 × 10^−6^	−4.0265	−0.7272	0.0127269
HHO	0.05173	0.3576	11.242	−7.48 × 10^−5^	−2.33 × 10^−4^	−4.0539	−0.7271	0.0126717
EGGO	0.05178	0.35884	11.1704	−2.06 × 10^−4^	−1.56 × 10^−4^	−4.0561	−0.7263	0.0126703

**Table 6 biomimetics-10-00545-t006:** Comparison of the best solution for gear train design problem.

Algorithm	Optimal Values for Variables				Optimum Cost
	*x* _1_	*x* _2_	*x* _3_	*x* _4_	
EGGO	43	19	16	49	2.700857 × 10^−12^
GMO	43	19	16	49	2.700857 × 10^−12^
KABC	50.4259	22.3987	16.7082	51.4394	0
IAPSO	43	19	16	49	2.700857 × 10^−12^
MBA	43	19	16	49	2.700857 × 10^−12^
ALO	43	19	16	49	2.7009 × 10^−12^

**Table 7 biomimetics-10-00545-t007:** Comparison of the best solution for three-bar truss design problem.

Algorithm	Optimal Values for Variables		Optimum Cost
	*x* _1_	*x* _2_	
EGGO	0.78868624	0.40823425	263.8958434
GMO	0.7886775	0.4082415	263.8958434
KABC	0.7886	0.4084	263.8959
DMMFO	0.788687421	0.408213541	263.8958435
GOA	0.7888976	0.4076196	263.895881
ALO	0.788662816000317	0.408283133832901	263.8958434
CS	0.78867	0.40902	263.9716
GSA	0.7886751284	0.4082483080	263.8958434
MBA	0.7885650	0.4085597	263.8958522

## Data Availability

The data that support the findings of this study are available from the corresponding author, upon reasonable request.

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
