# Peer review of "Novel Greylag Goose Optimization Algorithm with Evolutionary Game Theory (EGGO)"

_biomimetics, 2025, doi:10.3390/biomimetics10080545_

Round 1
Reviewer 1 Report
Comments and Suggestions for Authors
This paper presents a novel greylag goose optimization algorithm with evolutionary game theory (EGGO). The integration of evolutionary game theory provides a novel theoretical foundation, and the Lyapunov stability analysis strengthens the theoretical rigor. The extensive comparative studies and engineering applications demonstrate practical value. In my viewpoint, I think the theoretical foundation of this paper is solid, and the prosed experimental methodology is thorough, though some aspects of implementation clarity and parameter sensitivity analysis could be strengthened.
MY Detailed Review Comments are as follows.
1. I think the mathematical formulation and theoretical foundation need some improvements. For example, in this paper, the integration of evolutionary game theory into the GGO framework is conceptually sound, but several mathematical aspects require clarification. In Equation (2), you define the payoff matrix H with benefit functions h(s₁), h(s₂), and h(s₃), yet the specific calculation methodology for these benefit values remains unclear. How exactly do you map the fitness improvements achieved by each strategy to these mathematical formulations? Additionally, the transition from the continuous-time replicator dynamics in Equation (1) to the discrete implementation in your algorithm needs more explicit explanation. What discrete approximation scheme do you employ, and how does this affect convergence guarantees?
Besides, in this paper, the Lyapunov stability proof in Section 2.3 provides valuable theoretical grounding, but the assumption that ∂hₘ/∂yₘ ≤ 0 requires stronger justification. Under what specific conditions does this competitive relationship hold in your optimization context? Furthermore, the proposed proof demonstrates convergence to an ESS, but how do you ensure this ESS corresponds to a global optimum rather than a local equilibrium?
2. In terms of the algorithm implementation and parameter sensitivity, I think there are some improvements are needed. First, the pseudocode in Table 1 lacks sufficient detail regarding the dynamic parameter adaptation mechanisms. The strategy proportion updates based on Equation (1.1) - but this equation reference appears to be inconsistent with your main text numbering. How do you handle the initial strategy proportions, and what happens when strategy proportions approach boundary values (0 or 1)?
Second, the parameter z in Equation (9) decreases exponentially, but you don't specify the exponential decay rate or provide sensitivity analysis for this choice. Have you investigated how different decay rates affect algorithm performance? Similarly, the dynamic grouping mechanism adjusts n₁ and n₂ based on fitness distribution, but the specific criteria for these adjustments are not clearly articulated. What threshold values trigger group size modifications?
3. Your experimental comparison covers eight algorithms across multiple benchmark suites, which is commendable. However, the parameter settings in Table 2 appear to use default values from literature without adaptation to your specific test environment. Did you perform any parameter tuning for the comparison algorithms to ensure fair evaluation? This is particularly important since you've presumably optimized EGGO's parameters.
4. Given that EGGO maintains O(N•T•D) complexity similar to GGO, what accounts for the increased computational overhead? Is this trade-off between solution quality and computational cost acceptable for practical applications?
5. The engineering design problems demonstrate practical applicability, but your constraint handling approach is not explicitly described.
6. How does EGGO manage constraint violations during the evolutionary process?
7. Do you employ penalty functions, repair mechanisms, or constraint-preserving operators? This is crucial for understanding the algorithm's reliability in engineering contexts.
8. In the tension/compression spring design problem, your results show EGGO achieving slightly worse objective values than some competitors (e.g., WOA: f=0.0126712 vs EGGO: f=0.0126713). How do you reconcile this with claims of superior performance? Are these differences statistically significant?
9. How do you select individuals for local search, what local search operators do you employ, and how do you balance local search intensity with population-level exploration?
10. Based on your research focus on evolutionary game theory applications in optimization algorithms and strategic decision-making in complex systems, I recommend citing the following relevant works: Ref.[1]- doi:10.3390/math13030373. Concretely, Ref. [1] directly supports your evolutionary game theory foundation by demonstrating practical applications of game-theoretical approaches in complex optimization scenarios. The electricity market optimization context provides valuable insights into strategic behavior modeling that parallels your approach in EGGO. The authors' treatment of Nash equilibrium concepts and strategic stability analysis offers theoretical frameworks that could strengthen your ESS formulation and convergence analysis. Additionally, their multi-agent decision-making methodology provides relevant context for your dynamic strategy adjustment mechanisms. The authors' discussion of replicator dynamics and evolutionary stable strategies offers theoretical validation for your mathematical formulation in Equations (1)-(3). Their analysis of strategic adaptation mechanisms and population dynamics provides methodological insights that could enhance your dynamic grouping approach and strategy proportion updates. The paper's treatment of large-scale optimization challenges also supports your claims about scalability and high-dimensional problem solving capabilities.
Author Response
Thank you for your feedback. For detailed answers, please refer to the attached file.

Reviewer 2 Report
Comments and Suggestions for Authors
The Evolutionary Game Theory (EGT) is integrated into the GreyGoose Optimization (GGO) algorithm. The proposed algorithm is validated on case studies.
Contribution of the authors is pointed out at the end of the introduction
Remarks:
- Based on case studies performed the proposed EGGO algorithm has shown good accuracy and reasonable complexity. The execution time was in 1-st place in the case of 4 samples of 12 . Please add some comment in regard to complexity in comparison with other methods.
- What about implementation complexity? The proposed EGGO method has 9 parameters. Please describe evaluation/tuning of these parameters? Does it take reasonable time? or need extra experience?
- In engineering (study include engineering samples) is still dominating genetic algorithm (GA see doi: 10.5755/j02.ms.33259). Thus, comparison with GA should be at least discussed.
- The sample problems considered cover differentiable functions , thus the hybrid methods like (GA+gradient) can be applied? Not discussed?
Author Response
Thank you for your suggestion. The detailed reply can be found in the attachment.
